# Interpretation of Ambulatory Blood Pressure Monitoring for Risk Stratification in Hypertensive Patients: The ‘Ambulatory Does Prediction Valid (ADPV)’ Approach

**DOI:** 10.3390/diagnostics13091601

**Published:** 2023-04-30

**Authors:** Fabio Angeli, Gianpaolo Reboldi, Francesco Giuseppe Solano, Antonietta Prosciutto, Antonella Paolini, Martina Zappa, Claudia Bartolini, Andrea Santucci, Stefano Coiro, Paolo Verdecchia

**Affiliations:** 1Department of Medicine and Technological Innovation (DiMIT), University of Insubria, 21100 Varese, Italy; 2Department of Medicine and Cardiopulmonary Rehabilitation, Istituti Clinici Scientifici Maugeri IRCCS, 21049 Tradate, Italy; 3Department of Medicine, and Centro di Ricerca Clinica e Traslazionale (CERICLET), University of Perugia, 06100 Perugia, Italy; 4Division of Nephrology, Hospital S. Maria della Misericordia, 33100 Perugia, Italy; 5USL Umbria 1, 06127 Perugia, Italy; 6Department of Medicine and Surgery, University of Insubria, 21100 Varese, Italy; 7Division of Cardiology, Hospital S. Maria della Misericordia, 06100 Perugia, Italy; 8Fondazione Umbra Cuore e Ipertensione-ONLUS, 06100 Perugia, Italy

**Keywords:** hypertension, blood pressure, ambulatory blood pressure, ambulatory blood pressure monitoring, prognosis, epidemiology

## Abstract

Several outcome-based prospective investigations have provided solid data which support the prognostic value of 24 h ambulatory blood pressure over and beyond cardiovascular traditional risk factors. Average 24 h, daytime, and nighttime blood pressures are the principal components of the ambulatory blood pressure profile that have improved cardiovascular risk stratification beyond traditional risk factors. Furthermore, several additional ambulatory blood pressure measures have been investigated. The correct interpretation in clinical practice of ambulatory blood pressure monitoring needs a standardization of methods. Several algorithms for its clinical use have been proposed. Implementation of the results of ambulatory blood pressure monitoring in the management of individual subjects with the aim of improving risk stratification is challenging. We suggest that clinicians should focus attention on ambulatory blood pressure components which have been proven to act as the main independent predictors of outcome (average 24 h, daytime, and nighttime blood pressure, pulse pressure, dipping status, BP variability).

## 1. Introduction

Since diagnosis and clinical management of hypertension are based on blood pressure (BP) measurements taken in the physician’s office, most of the diagnostic and treatment recommendations issued by major hypertension guidelines are based on office BP [1,2]. Nonetheless, 24 h non-invasive ambulatory BP monitoring (ABPM) is increasingly used to refine cardiovascular risk stratification [3,4,5,6,7,8,9,10]. It is today well established that BP measured in the hospital or in the doctor’s office (‘office’ or ‘clinic’ BP) generally provides limited information on the real BP load and variability over the 24 h period. On the other hand, more frequent ambulatory BP measurements during the entire 24 h period can provide an accurate landscape of the real patient’s BP during usual daily activities and during sleep. As depicted in Figure 1, at any level of office BP, the observed ambulatory BP varies dramatically around the value predicted from the regression equation.

The independent association between ambulatory BP, hypertensive target organ damage, and the risk of cardiovascular events in hypertensive patients (either treated or untreated) is now well established [3,4,11,12,13]. The main aim of our narrative review is to analyze the advantages of ambulatory BP over office BP in refining risk stratification and in predicting clinical outcomes in hypertension. We also discussed the most appropriate way of interpreting ambulatory BP profiles in the prognostic evaluation of patients with hypertension.

## 2. Average Ambulatory BP

ABPM enables clinicians to obtain a precise estimation of a patient’s BP profile by measuring BP during daytime and nighttime periods [14]. Solid evidence from large longitudinal clinical studies supports the prognostic impact of average ambulatory BP over and beyond office-based BP [14].

Some landmark studies have addressed the independent prognostic value of average ambulatory BP. In the Ohasama study, ambulatory BP predicted the risk of cardiovascular mortality after adjustment for several potential confounders, including office BP [15]. An analysis of a Danish population demonstrated that average ambulatory daytime systolic BP was a stronger predictor of adverse outcomes independent of office-based systolic BP [16,17]. An analysis of the International Database on Ambulatory blood pressure in relation to Cardiovascular Outcome (IDACO) study, conducted in 7030 subjects from Denmark, Belgium, Japan and Sweden [17], concluded that average daytime ambulatory BP was a stronger predictor for the risk of cardiovascular events than office-based BP measurement. In a fully-adjusted model which included both ambulatory and office BP, office BP was no longer statistically significant, while daytime systolic BP conversely remained strongly significant [17]. Similar results were documented in elderly subjects with elevated systolic and normal or low diastolic BP (‘isolated systolic hypertension’). Ambulatory systolic BP proved to be a stronger predictor of cerebrovascular and cardiovascular events when compared with office BP [18]. Studies conducted in independent centers with similar experimental design provided similar results (Table 1).

### 2.1. White Coat Hypertension

Numerous clinical studies and guidelines have defined white coat hypertension (WCH), also referred to as ‘isolated office hypertension’, as a phenotype characterized by hypertension detected by the doctor in his/her office in individuals with normal BP at home or during normal daily activities [26,27,28,29]. The prognostic significance of WCH has been investigated in many outcome-based studies. Some cohort and interventional studies showed that cardiovascular risk in subjects with WCH (a) is comparable to that of clinically normotensive subjects and (b) is consistently lower than in subjects with elevated ambulatory BP [17]. Other studies provided different results by showing that organ damage and risk of events are slightly increased in subjects with WCH than in clinically normotensive subjects [29,30].

In a study published in *The Lancet*, we subdivided hypertensive patients with WCH into two groups with pretty low (<130/80 mmHg) or intermediate (between 130/80 and 131/86 mmHg in women or 136/87 mmHg in men) daytime ambulatory BP. The difference in the risk of major cardiovascular events between clinically normotensive subjects and the hypertensive group with WCH defined restrictively was not statistically significant. Conversely, the differences between the normotensive group and the WCH group defined less restrictively were significant. In other words, these results suggest that a daytime ambulatory BP < 130/80 mmHg identifies hypertensive patients with WCH who are really at low cardiovascular risk. This analysis has been more recently replicated in a huger sample of 3174 patients with 376 major cardiovascular events over a mean follow-up period of 7 years. The analysis confirmed that the clinically normotensive group and the group of patients with WCH defined more restrictively (daytime BP < 130/80 mmHg) were comparable in terms of their risk of major cardiovascular disease. Conversely, the differences between the clinically normotensive group and the WCH group defined less restrictively achieved significance. Notably, these data have been confirmed in independent prospective studies [31,32] which included 1038 individuals with mild hypertension followed for an average of 4.5 years [31] and 958 elderly Japanese patients followed for an average of 3.5 years [32].

An international collaborative clinical study collected individual patient data from 4406 patients from four prospective cohort studies conducted in Italy, the United States, and Japan who were followed for a median of five years [10]. In a fully adjusted multivariable model, the hazard ratio for stroke was 1.15 (0.61–2.16) in the WCH group and 2.01 (1.31–3.08) in the ambulatory hypertension group compared with the normotensive group [10]. Nonetheless, stroke risk increased in the WCH group over time and the Kaplan–Meier curve in the WCH group crossed that of the ambulatory hypertension group by approximately the ninth year of follow-up. Based on these data, we suggested that WCH might not be a fully benign condition for stroke in the very long term [10].

Consequently, hypertension guidelines suggest that subjects with WCH should be extensively investigated for concomitant risk factors and target organ damage in order to inform therapeutic decisions [1]. So far, there are no randomized studies comparing a ‘drug treatment’ with ‘no treatment’ in subjects with WCH in the absence of other indications for treatment [1].

### 2.2. Masked Hypertension

Over the years, several studies have addressed a new clinical phenotype with normal BP (i.e., <140/90 mmHg) in the doctor’s office and increased BP levels during usual daily activities outside the doctor’s office. Pickering defined this condition as ‘masked hypertension’ (MH). Clearly, the term MH underscores the concept that hypertension might not be diagnosed in these individuals on the basis of office BP measurements. Various factors are known to increase BP during usual daily life (alcohol, physical activity, diabetes, obesity, psychological stress, cigarette smoking, etc.) and these factors could thus raise BP out of the doctor’s office.

Several studies have shown that hypertensive organ damage is comparable between patients with MH and those with elevated BP during normal daily activities, despite the remarkable differences in office BP. From a prognostic standpoint, patients with MH showed a 1.5- to 3-fold higher risk of major cardiovascular disease than subjects with normotension, and their outcomes did not differ significantly from those of patients with elevated office and out-of-office BP [5,33,34,35]. Similarly, MH defined by self-measured home BP proved to have prognostic significance [5,36]. In a study conducted in 4939 treated hypertensive patients (mean age 70 years) followed for about 3 years, the cardiovascular disease rate was only 11.1/1000/year when office BP was <140/90 and self-measured BP at home was <135/85, while this rate was 30.6/1000/year in the presence of MH, defined by a home BP ≥ 135/ 85 mmHg and an office BP < 140/90 [36].

In a meta-analysis, we evaluated the prognostic impact of MH defined by either ambulatory or self-measured BP [5]. We examined data from prospective studies, which included both normotensive and hypertensive subjects, either treated or untreated, and MH defined by ambulatory BP or self-measured BP [5]. We calculated the hazard ratio (HR) and 95% confidence interval (CI) for each study separately, and then derived the HR and 95% CI according to random-effects models [5]. Based on the aforementioned criteria, we identified eight clinical studies divided in two subgroups: (a) studies which investigated the prognosis of MH detected by ambulatory BP (*n* = 6) and (b) studies which investigated the prognostic impact of MH detected by self-measured BP (*n* = 2) [5]. The risk of major cardiovascular disease events was considerably higher in the subjects with MH than in the normotensive subjects, regardless of whether the definition of MH was based on self-measured BP (HR: 2.13; 95% CI: 1.35–3.35; *p* = 0.001) or 24 h ABPM (HR: 2.00; 95% CI: 1.54–2.60; *p* < 0.001) [5].

Overall, these data strongly suggest that MH should be considered an insidious and prognostically adverse condition that can be reliably diagnosed by self-measured BP and ambulatory BP. Antihypertensive treatment is advised in these subjects, although the associated outcome benefits are still undetermined. So far, there are no randomized outcome-based studies specifically designed to assess the prognostic impact of hypertensive treatment in subjects with MH.

Although average ambulatory BP (24 h, daytime, and nighttime BP) has been extensively investigated as a prognostic determinant in hypertension, other measures exist for describing different aspects of ambulatory readings. They include changes from day to night in BP, morning surges, pulse pressure (PP), and estimates of BP variability [37,38,39,40,41,42,43,44,45].

## 3. Day–Night BP Changes and Early Morning Rises in BP

It is now well established that there is a physiological decline in BP from the daytime to the nighttime window, with BP values tending to peak during the awake period and then falling to a nadir during the night [46,47]. In this context, ABPM is the best tool to investigate BP changes from day to night [48].

### 3.1. Dipping Status

The “dipper/non-dipper” classification was first introduced by O’Brien in 1988 on the basis of clinical observations of a more frequent history of stroke among subjects with absent or blunted BP decline from day to night [49].

Subjects with a day–night BP difference below a given value are usually referred to as ‘non-dippers’, and the remaining subjects are usually referred to as ‘dippers’. To date, the suggested values for this distinction range from 10% to 10/5 mmHg up to 0% (i.e., no reduction in BP from day to night or higher BP during the night than during the day) [50]. Furthermore, BP in some patients is actually higher during the night than during the day (Figure 2). This phenomenon is usually referred to as ‘reverse dipping’ and is associated with an adverse cardiovascular outcome [51].

After evidence of an association between “non-dipping” patterns and target organ damage (including left ventricular hypertrophy, microalbuminuria, angiographic coronary artery stenosis, and previous cerebrovascular events with or without cognitive dysfunction [41,49,52,53,54]), several outcome-based studies confirmed that a diminished nocturnal BP fall is associated with poor prognosis in hypertensive subjects [55], thereby refining cardiovascular risk stratification (Table 2) [8,56,57,58,59].

Several factors, including chronic kidney disease (CKD) and left ventricular hypertrophy (LVH), may influence the degree of BP fall from day to night. Consequently, the prognostic impact of the day–night BP fall could be influenced by CKD and LVH. In a recent analysis of the PIUMA study, the day–night BP fall remained a predictor of cardiovascular risk in the absence of data regarding CKD or LVH. However, somewhat unexpectedly, after adjustment for CKD (estimated glomerular filtration rate <60 mL/min or proteinuria) and LVH, the day–night BP fall was no longer significant for risk stratification in these patients [58]. These data strongly suggest that a non-dipping pattern is a sort of proxy of hypertensive organ damage at renal or cardial level [58]. The day–night BP fall did not improve the prognostic information provided by CKD or LVH. Interestingly, the independent competing models which included 24 h, daytime or nighttime BP combined with the day–night BP fall were equally informative for cardiovascular risk prediction. In other words, when clinical data on CKD and LVH are available, the day–night BP fall is of secondary clinical importance.

### 3.2. Early Morning BP Surge

More recently, some studies have analyzed the association between early morning BP surges and outcomes in hypertension. It is well known that myocardial infarction, stroke, and sudden cardiac death tend to cluster in the early morning, especially during the 4–6 h after awakening [51].

These observations support the hypothesis of a pathophysiological relationship between hemodynamic aberrations in the early morning (Figure 3) and vascular damage [62,63,64,65]. On the other hand, the hypothesis of an adverse prognostic significance of a blunted or reversed diurnal BP rhythm, demonstrated in several clinical studies, may be difficult to reconcile with the assumption that an excessive early morning rise in BP is also predictive of a worse outcome. Indeed, an exaggerated early morning rise in BP could be expected more frequently among dippers than non-dippers or reverse dippers. Unfortunately, most studies that have evaluated the morning BP surge have been conducted in relatively small cohorts of individuals with high BP who were untreated at the time of 24 h monitoring. The impact of antihypertensive treatment on the day–night BP fall and the early morning rise in BP could indeed be critical.

Different estimates of the morning surge in BP have been proposed [42,66], including the sleep-trough morning BP surge (difference between the average BP during the 2 h following awakening and the lowest nighttime BP) and the pre-awakening morning BP surge (difference between the average BP during the 2 h after awakening and the average BP during the 2 h before awakening).

The early morning BP surge has been linked with vascular disease [67], LVH [68], brain lesions [42]), stroke [42], and a composite pool of cardiovascular events [66]. However, contrary to expectations, we found in a recent analysis of the PIUMA study [56] that an excessive early morning rise in BP was not an independent predictor of increased cardiovascular risk in hypertension. The same analysis found a link between the day–night dip and the morning BP surge [56]. Similarly, the Pressioni Arteriose Monitorate E Loro Associazioni (PAMELA) study provided evidence that the magnitude of the morning BP rise is not associated with the risk of cardiovascular outcomes [69]. In a clinical study by Pierdomenico et al., a high early morning surge was linked with an increased risk of stroke in dippers, but not in non-dippers, among elderly patients with hypertension [70].

## 4. Blood Pressure Variability

Ambulatory BP variability is traditionally assessed from the standard deviation (SD) of BP measurements over 24 h (or more appropriately, daytime and nighttime periods evaluated separately). However, other indices of BP variability have been proposed [71], including the ‘weighted’ SD of the 24 h mean value [72], BP variability independent of the mean (VIM) [73], maximum minus minimum BP (MMD) [71,74], and variability ratios [75].

Ambulatory BP variability has been extensively investigated in hypertension as a significant determinant of prognosis [13,26,76,77]. Abnormal ambulatory BP variability has been linked to hypertensive target organ damage [77,78,79,80], cardiovascular outcomes [81,82,83,84], and all-cause death [84,85]. In this setting, an analysis from our group showed that increased variability in systolic BP during the nighttime (defined by SD > 10.8 mmHg) was an independent predictor of cardiac events in initially untreated hypertensive subjects [86]. In this study, 2649 initially untreated hypertensive subjects were followed for up to 16 years (mean follow-up 6 years) [86]. We used SD of daytime and nighttime BP to estimate variability in BP. The group median of SD identified patients with low or high BP variability (13/10 mmHg for daytime systolic/diastolic BP; 11/9 mmHg for nighttime systolic/diastolic BP). There were 167 new cardiac events and 122 new cerebrovascular events at follow-up [86]. The patients with higher BP variability showed a higher rate of cardiac events when compared with those with lower variability, and comparable results were noted for cerebrovascular events. In a fully adjusted multivariable model, increased systolic BP variability during the night was associated with a 51% higher risk of cardiac events [86].

A recent consensus document of the European Society of Hypertension synthetizes current evidence, clinical implications, and unmet needs in the setting of BP variability [76].

## 5. Pulse Pressure

Pulse pressure (PP) is a well-recognized marker of arterial stiffness. A fundamental mechanism underlying the rise in PP with aging is the stiffening of large arteries. An increased PP is thus a marker of stiff arterial walls, with several adverse implications of potential prognostic value [87]. We found a direct association between elevated neutrophil count, a marker of systemic inflammation, and 24 h ambulatory PP among postmenopausal women with hypertension. Of note, this association remained significant after adjusting for several confounders, including age, serum glucose, and left ventricular hypertrophy [88]. Other studies found an association between PP and carotid atherosclerosis, left ventricular mass, and white matter lesions.

A link between PP and major cardiovascular events has been reported in several studies, and the link was independent of systolic BP and diastolic BP [89,90]. Ambulatory PP was more potent that office PP for risk stratification in hypertensive subjects [43]. This information emerged from a study conducted in 2010 initially untreated and uncomplicated subjects with essential hypertension [43]. The rates of total cardiovascular events per 100 patient-years in the three tertiles (≤45 mmHg, 46–53 mmHg, and >53 mmHg) of the distribution of average 24 h PP were 1.19, 1.81, and 4.92 (Figure 4). Rates of fatal events were 0.11, 0.17, and 1.23 (*p* < 0.01) [43]. Similar results have been documented for office PP (Figure 4). However, after allowances for other risk factors, cardiovascular morbidity and mortality was better predicted by ambulatory PP than by office PP, even after controlling for multiple risk factors [43].

In a study conducted in postmenopausal hypertensive women [88], we found that high values for 24 h ambulatory PP are strong predictors of adverse outcomes. Specifically, the rate of cardiovascular events was 1.02, 1.36, and 3.75 per 100 patient-years in the three tertiles (<48 mmHg, 48–56 mmHg, and ≥57 mmHg) of the distribution of 24 h PP (*p* < 0.0001). In a multivariable analysis, 24 h ambulatory PP was an independent predictor of total cardiovascular events. For each 10 mmHg increase in 24 h PP, the risk of total cardiovascular events increased by 73% after controlling for the influence of other risk markers, including age, diabetes, serum creatinine, HDL cholesterol, LDL cholesterol, total white blood cell count, and left ventricular hypertrophy. Notably, office-recorded PP did not achieve significance when used in the same model [88].

Overall, these data strongly suggest that 24 h PP is a strong predictor of major cardiovascular disease events in hypertensive patients.

To further characterize the clinical impact of PP, we also analyzed the different prognostic impact of 24 h mean BP and PP on stroke and coronary artery disease [91]. We analyzed 2311 subjects with essential hypertension prospectively followed for up to 14 years [91]. After allowance for sex, age, and traditional cardiovascular risk factors, for each 10 mmHg increase in 24 h PP, the risk of cardiac events increased by 35% [91]. Unexpectedly, 24 h mean BP was not a significant predictor of cardiac events after controlling for PP [91]. On the other hand, for each 10 mmHg increase in 24 h mean BP, the risk of cerebrovascular events increased by 42%, whereas 24 h PP did not yield significance [91]. Thus, 24 h PP was an independent predictor of cardiac events, and 24 h mean BP was an independent predictor of cerebrovascular events.

These findings strongly suggest that ambulatory mean BP and PP exert a different predictive effect on cardiac and cerebrovascular outcomes [91]. While PP is the dominant predictor of cardiac events, mean BP is the major independent predictor of cerebrovascular events [91].

## 6. Conclusions

The available evidence supporting the prognostic value of ABPM is remarkable and soundly based on the outcome of prospective clinical studies [92]. ABPM should be considered to optimize the management of hypertensive patients through refinement of cardiovascular risk stratification above and beyond traditional risk markers and office BP. Although another approach to out-of-office BP measurement (i.e., home BP) may be used to provide a better prediction of organ damage and the risk of cardiovascular complications when compared with office BP [93], ABPM should not be replaced by home BP because of its unequivocal superiority under several diagnostic and prognostic aspects [93]. Home BP and ABPM are complementary techniques which should be used with the precise aim of exploiting the best that each technique can provide. ABPM is most appropriate for exploring BP at night and in specific conditions during the day (at work, in smokers, etc.). However, it is more expensive that home BP measurements and cannot be replicated many times. Conversely, self-measured home BP is ideal for long-term monitoring and is advised in almost all patients with hypertension. ABPM should be strongly considered in patients newly diagnosed with hypertension and not yet treated because most of the available outcome-based studies with ABPM have been conducted with this kind of patient. The main limitation of ABPM and home BP is the lack of randomized outcome-based studies comparing a therapeutic regimen based on the above techniques with a regimen based on traditional office BP for the prevention of cardiovascular disease.

The European Society of Hypertension (ESH) guidelines remark the concept that ABPM and self-measured home BP are valuable alternative to traditional office BP measurements as it may enable the diagnosis of WCH and MH, which would not be possible using office BP alone. ABPM has advantages over home BP in terms of enabling multiple nighttime BP readings, its possible use in real-life settings (work place, etc.), and more extensive evidence from the literature supporting it [1]. Self-measured home BP has the advantages of allowing BP monitoring for longer periods, although with fewer measurements per unit of time, and being relatively less expensive and more widely available [1].

Average 24 h, daytime, and nighttime BP remain the main components of the ambulatory BP profile proven to be prognostically significant (Figure 5). Other ambulatory measures are being investigated, including sophisticated estimations of BP variability, day–night BP changes, and morning BP surges. Their correct implementation in clinical practice of ABPM needs a standardization of methods.

Several algorithms for the clinical use of ABPM have been proposed. However, in our opinion, the interpretation of ABPM should be simplified in individual patients to optimize their management. We should probably focus on those components of ABPM which more clearly improved risk stratification in long-term outcome studies:-Average ambulatory BP;-Pulse pressure;-Dipping status;-Variability.

We suggest the use of the mnemonic ‘Ambulatory Does Prediction Valid’ (**A-D-P-V**: **A**verage ambulatory BP; **D**ipping pattern; **P**ulse pressure; **V**ariability in ambulatory BP) to correctly refine cardiovascular risk stratification and adapt treatment strategies (Figure 5). In subjects with office BP ≥ 140/90 mmHg, 24 h ambulatory BP identifies low-risk individuals with normal or optimal values (<130/80 mmHg) of daytime ambulatory BP (i.e., white coat hypertension without comorbidities and target organ damage) [6]. Conversely, increased 24 h PP (>53 mmHg), a non-dipping BP pattern (day–night BP reduction < 10%), or increased nighttime systolic BP variability (SD > 10.8 mmHg) in subjects with elevated daytime BP identifies patients at increased cardiovascular risk, regardless of office BP measurements [5].

## Figures and Tables

**Figure 1 diagnostics-13-01601-f001:**
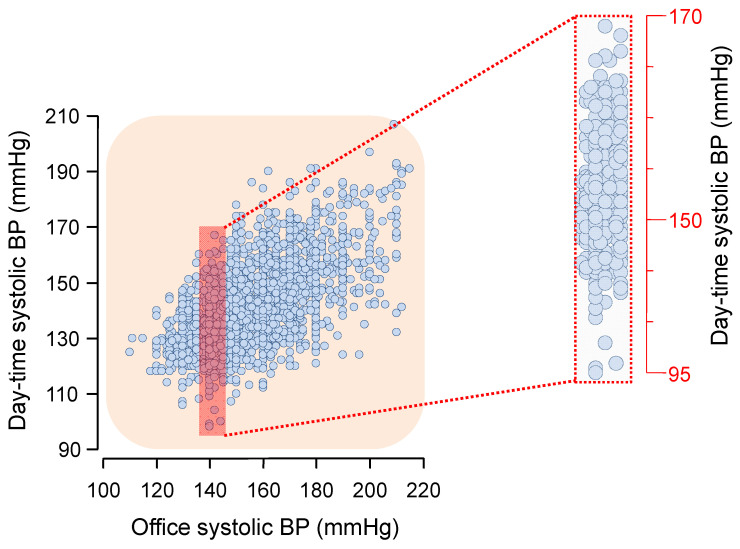
Association between ambulatory and office BP (see text for details). BP = blood pressure. Data from the PIUMA (Progetto Ipertensione Umbria Monitoraggio Ambulatoriale) study.

**Figure 2 diagnostics-13-01601-f002:**
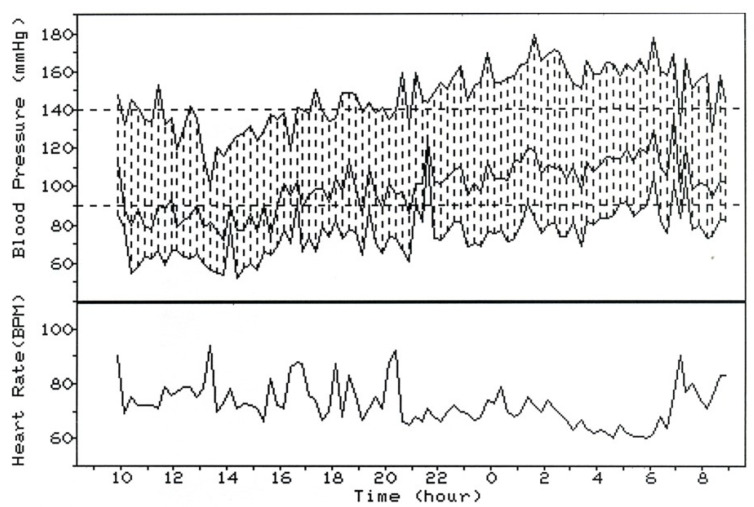
Higher BP during the night than during the day (‘reverse dipper’ or ‘riser’ profile) in a patient with chronic kidney disease. Data from the PIUMA (Progetto Ipertensione Umbria Monitoraggio Ambulatoriale) study.

**Figure 3 diagnostics-13-01601-f003:**
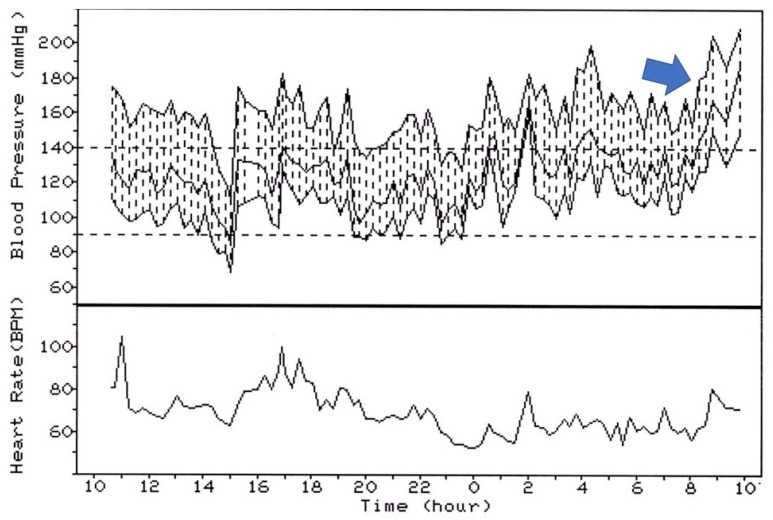
Early morning surge in blood pressure (arrow) in a patient who was not taking antihypertensive drugs during 24 h monitoring. Data from the PIUMA (Progetto Ipertensione Umbria Monitoraggio Ambulatoriale) study.

**Figure 4 diagnostics-13-01601-f004:**
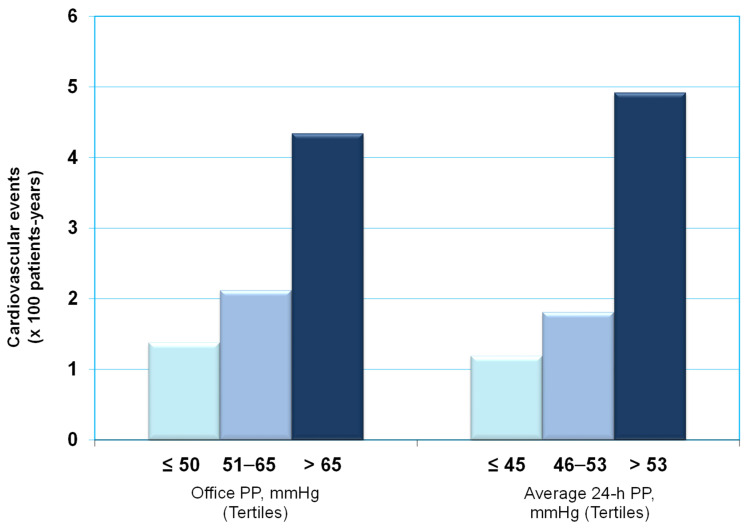
Association between ambulatory/office PP and the risk of cardiovascular events (see text for details). PP = pulse pressure.

**Figure 5 diagnostics-13-01601-f005:**
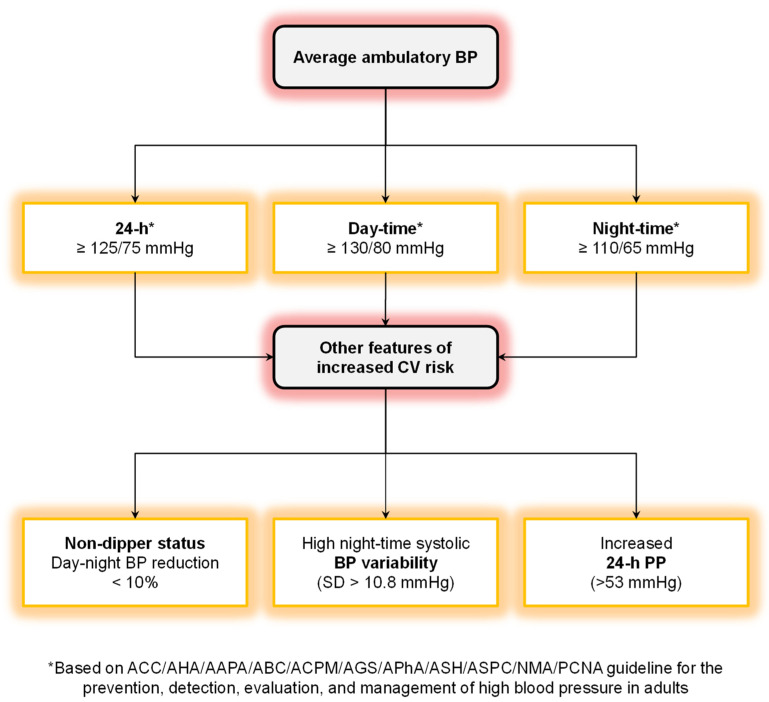
Association between ambulatory/office PP and the risk of cardiovascular events (see text for details). PP = pulse pressure. From ref. [2].

**Table 1 diagnostics-13-01601-t001:** Landmark outcome-based studies which assessed the prognostic significance of ambulatory BP (see text for details). PIUMA = Progetto Ipertensione Umbria Monitoraggio Ambulatoriale; OvA = Office Versus Ambulatory Blood Pressure; PAMELA = Pressioni Arteriose Monitorate E Loro Associazioni.

Study/Setting (N, Number of Patients)	Results	Ref.
Resistant hypertension (N = 86)	The cardiovascular event rate was significantly higher in the upper tertile than in the middle/lower tertiles of daytime diastolic BP.	[19]
PIUMA study; clinically hypertensive patients and normotensive controls (N = 2010)	The rate of major cardiovascular events was comparable in hypertensive patients with white coat hypertension and normotensive subjects (i.e., office BP < 140/90 mmHg). Among clinically hypertensive subjects it was higher in non-dippers than in dippers.	[20]
PIUMA study; clinically hypertensive patients and normotensive controls (N = 790)	The rate of major cardiovascular events was lower in hypertensive patients with ambulatory BP control (<135/85 mmHg: 0.71 events/100 person-years) than in uncontrolled patients (1.8 events/100 person-years). Ambulatory BP control, but not office BP control, was an independent predictor of a reduced risk of subsequent cardiovascular events.	[21]
OvA study; treated hypertensive patients (N = 1963)	In a multivariate model adjusted for several covariates, including office BP, average 24 h ambulatory BP was an independent predictor of major cardiovascular disease.	[22]
Dublin outcome study; patients referred for management of cardiovascular risk (N = 5292)	Ambulatory BP proved strongly superior to office BP in the prediction of cardiovascular mortality before and after allowances for confounders. In particular, elevated nighttime BP proved to be the most powerful independent prediction of poor outcomes.	[23]
PAMELA study; hypertensive patients (N = 2051)	Nighttime ambulatory BP was superior to daytime BP as a prognostic predictor.	[24]
Pierdomenico et al. (N = 738)	Patients with non-resistant and resistant masked uncontrolled hypertension are at increased risk of major cardiovascular events compared to subjects with controlled hypertension.	[25]

**Table 2 diagnostics-13-01601-t002:** Clinical studies assessing the prognostic value of non-dipping patterns (see text for details). PIUMA = Progetto Ipertensione Umbria Monitoraggio Ambulatoriale; Syst-Eur = Systolic Hypertension in Europe.

Study (N, Number of Patients)	Results	Ref.
Yamamoto et al. (N = 105)	The day–night BP reduction was lower in patients with future cerebrovascular events than in those without.	[60]
Syst-Eur trial (N = 808)	At any level of 24 h ambulatory BP, the risk of major cardiovascular disease increased with the night-to-day ratio of systolic BP.	[18]
Ohkubo et al. (N = 1542)	The rate of cardiovascular mortality was higher in ‘non-dippers’ and ‘reverse dippers’ than in ‘dippers’.	[61]
PIUMA study (N = 2934)	Regardless of the definition of day and night (i.e., fixed time intervals or based on a diary) the day–night BP changes are prognostically important.	[8]
PIUMA study (N = 2934)	The prognostic value of day–night BP drops is absent in patients who experience important alterations in the reported duration of sleep during nocturnal monitoring.	[7]

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
