# Peer review of "Interpretation of Ambulatory Blood Pressure Monitoring for Risk Stratification in Hypertensive Patients: The ‘Ambulatory Does Prediction Valid (ADPV)’ Approach"

_diagnostics, 2023, doi:10.3390/diagnostics13091601_

Round 1
Reviewer 1 Report
This paper reviewed clinical studies of ambulatory blood pressure (BP) monitoring over traditional BP monitoring, office BP. These studies highlight the advantage of ambulatory BP in predicting cardiovascular outcomes. In conclusion, this paper suggests that clinicians should pay more attention to ambulatory blood pressure.
Introduction
The second paragraph ‘The independent association between…’ is a sentence, not a paragraph, you can either combine it with the first paragraph or add more sentences to it.
Average ambulatory BP
Consultation with a language professional is highly suggested. There are grammar issues in multiple places in this paper. For example, ‘As aforementioned, ABPM enables clinicians to obtain a more precise estimation of a patient’s BP profile, assessing average… There is now solid evidences …’ ‘more’ may not be needed without ‘than’; ‘, assessing’ can be ‘and assess’; ‘There is’ should be ‘There are’.
Several clinical studies and results have been listed in Table 1 and 2. The number of subjects is also very important when a conclusion is made. Please add the number of subjects of each study in Table 1 and 2.
Early morning BP surge
The last paragraph is also a sentence. Please modify it.
Clinical studies included in this paper suggest that ambulatory BP monitoring is very valuable for a patient. However, to my understanding, continuous monitoring of BP is very challenging, I cannot imagine I monitor my BP every single hour. As a review paper, I suggest discussing the disadvantage of ambulatory BP. Besides, this paper briefly mentioned self-measured BP. I would suggest this paper also should overview the comparison of ambulatory and self-measured BP with sufficient citations. Furthermore, I think it is very important to see the author’s opinion, comments, and suggest for using ambulatory, office and/or self-measured BP in the practical world.
Author Response
To Reviewer 1
You wrote: This paper reviewed clinical studies of ambulatory blood pressure (BP) monitoring over traditional BP monitoring, office BP. These studies highlight the advantage of ambulatory BP in predicting cardiovascular outcomes. In conclusion, this paper suggests that clinicians should pay more attention to ambulatory blood pressure.
Response: thank you for your helpful comments. Changes in the text are marked in red.
You wrote: Introduction. The second paragraph ‘The independent association between…’ is a sentence, not a paragraph, you can either combine it with the first paragraph or add more sentences to it.
Response: we modified the text. Thank you.
You wrote: Average ambulatory BP. Consultation with a language professional is highly suggested. There are grammar issues in multiple places in this paper. For example, ‘As aforementioned, ABPM enables clinicians to obtain a more precise estimation of a patient’s BP profile, assessing average… There is now solid evidences …’ ‘more’ may not be needed without ‘than’; ‘, assessing’ can be ‘and assess’; ‘There is’ should be ‘There are’.
Response: done. Thank you.
You wrote: Several clinical studies and results have been listed in Table 1 and 2. The number of subjects is also very important when a conclusion is made. Please add the number of subjects of each study in Table 1 and 2.
Response: done. Thank you.
You wrote: Early morning BP surge. The last paragraph is also a sentence. Please modify it.
Response: we modified the text. Thank you.
You wrote: Clinical studies included in this paper suggest that ambulatory BP monitoring is very valuable for a patient. However, to my understanding, continuous monitoring of BP is very challenging, I cannot imagine I monitor my BP every single hour. As a review paper, I suggest discussing the disadvantage of ambulatory BP. Besides, this paper briefly mentioned self-measured BP. I would suggest this paper also should overview the comparison of ambulatory and self-measured BP with sufficient citations. Furthermore, I think it is very important to see the author’s opinion, comments, and suggest for using ambulatory, office and/or self-measured BP in the practical world.
Response: we agree. We modified conclusions to highlight advantages of ABPM. Specifically, we stated (page 12): “Although another approach to out-of-office BP measurement (i.e. home BP) may be used to provide a better prediction of organ damage and the risk of cardiovascular complications when compared with office BP [112,113], ABPM should not be replaced by home BP because of its unequivocal superiority under several diagnostic and prognostic aspects [112,113]. Home BP and ABPM are complementary techniques which should be used with the precise aim of exploiting the best that each technique can provide [113]. ABPM is most appropriate for exploring BP at night and in specific conditions during day (i.e., at work, in smokers, etc). However, it is more expensive that home BP and cannot be replicated many times. Conversely, self-measured home BP is ideal for long-term monitoring and it is adviced in almost all patients with hypertension. ABPM should be strongly considered in patients newly diagnosed with hypertension and not yet treated because most of the available outcome-based studies with ABPM have been conducted in this kind of patients. The main limitation of ABPM and home BP is the the lack of randomized outcome-based studies comparing a therapeutic regimen based on the above techniques with a regimen besed on traditional office BP for prevention of cardiovascular disease.”.
Reviewer 2 Report
Recommendation: Need minor revision.
This review article provides a comprehensive summary of the key components of ambulatory blood pressure that clinicians should focus on as independent predictors of outcome. The authors delve into details about average 24-hour, day-time and night-time blood pressure, pulse pressure, dipping status, and BP variability. Overall, the article is well-written and provides updated references, although minor revisions to the format could enhance readability. Please see details below and they should be easily addressed.
1. P.2 line 59, should be “evidence”.
2. P.2 line 61, change “addressed” to “have addressed”?
3. P.3 line 81 Table 1, “in the middle/lower tertile of daytime diastolic BP”, missing a period at the end?
4. P.4 line 118, a typo error, “appoximately” should be “approximately”.
5. P.5 line 133, change “damages” to “damage”.
6. P.5 line 139, “proved to have prognostic significance” may be better?
7. P.5 line 160, change “so fare” to “so far”.
8. P.6 line 194, a typo error, should be “without”.
9. P.6 line 196, should be “thereby”.
10. P.7 line 206, a typo error, change “bt” to “by”.
11. P.7 line 214, a typo error, should be “night-time”.
12. P.8 line 231, remove the period in “day-night”.
13. P.8 line 240, should be “night-time”.
14. P.9 line 270, a typo error, should be “identified”.
15. P.9 line 274, a typo error, should be “compared”.
16. P.12 line 341, “above and beyond”?
17. P.12 line 348, a typo error, should be “multiple”.
18. P.12 line 360, should be “simplified”.
19. P.12 line 372, should be “comorbidities”.
Author Response
To Reviewer 2
You wrote: Recommendation: Need minor revision. This review article provides a comprehensive summary of the key components of ambulatory blood pressure that clinicians should focus on as independent predictors of outcome. The authors delve into details about average 24-hour, day-time and night-time blood pressure, pulse pressure, dipping status, and BP variability. Overall, the article is well-written and provides updated references, although minor revisions to the format could enhance readability. Please see details below and they should be easily addressed.
Response: thank you for your helpful comments. Changes in the text are marked in red.
You wrote: 1. P.2 line 59, should be “evidence”. 2. P.2 line 61, change “addressed” to “have addressed”? 3. P.3 line 81 Table 1, “in the middle/lower tertile of daytime diastolic BP”, missing a period at the end? 4. P.4 line 118, a typo error, “appoximately” should be “approximately”. 5. P.5 line 133, change “damages” to “damage”. 6. P.5 line 139, “proved to have prognostic significance” may be better? 7. P.5 line 160, change “so fare” to “so far”. 8. P.6 line 194, a typo error, should be “without”. 9. P.6 line 196, should be “thereby”. 10. P.7 line 206, a typo error, change “bt” to “by”.
- P.7 line 214, a typo error, should be “night-time”. 12. P.8 line 231, remove the period in “day-night”. 13. P.8 line 240, should be “night-time”. 14. P.9 line 270, a typo error, should be “identified”. 15. P.9 line 274, a typo error, should be “compared”. 16. P.12 line 341, “above and beyond”? 17. P.12 line 348, a typo error, should be “multiple”. 18. P.12 line 360, should be “simplified”. 19. P.12 line 372, should be “comorbidities”.
Response: we carefully revised the manuscript and we modified the text according to your suggestions. Thank you.
Reviewer 3 Report
This is an interesting review by Angeli et al.
However, some concerns are listed below.
Lines 292-302: a better characterization of the study group (the blood pressure and pulse pressure values in the 3 tertiles) is required.
“At any 300 given level of office PP, cardiovascular morbidity and mortality increased with 24-hour PP [61].”- this sentence is unclear
Lines 303-306: a better characterization of the study group (the pulse pressure values in the 3 tertiles) is required.
Several spelling mistakes can be identified in the document.
I feel that wherever a study conducted by this team is cited, for example the pulse pressure paragraph, other studies conducted by other teams on the same topic (for instance prognostic impact of 24-hour mean BP and PP on stroke and coronary artery disease) should be included.
Can the authors provide cutoff values for the proposed parameters(A-D-P-V) in the conclusion section from which adcerse cardiovascular or cerebrovascular events and end-organ hypertensive dammage are more likely?
An impressive number of studies conducted by this team is cited. Are all these self citations relevant and do they add to the value of this review?
Author Response
To Reviewer 3
You wrote: This is an interesting review by Angeli et al. However, some concerns are listed below.
Response: thank you for your helpful comments. Changes in the text are marked in red.
You wrote: Lines 292-302: a better characterization of the study group (the blood pressure and pulse pressure values in the 3 tertiles) is required.
Response: done. Thank you.
You wrote: “At any 300 given level of office PP, cardiovascular morbidity and mortality increased with 24-hour PP [61].”- this sentence is unclear
Response: we’re sorry for the lack of clarity. We modified the text accordingly. Thank you.
You wrote: Lines 303-306: a better characterization of the study group (the pulse pressure values in the 3 tertiles) is required.
Response: done. Thank you.
You wrote: Several spelling mistakes can be identified in the document.
Response: we carefully revised the manuscript. Thank you.
You wrote: I feel that wherever a study conducted by this team is cited, for example the pulse pressure paragraph, other studies conducted by other teams on the same topic (for instance prognostic impact of 24-hour mean BP and PP on stroke and coronary artery disease) should be included.
Response: done. Thank you.
You wrote: Can the authors provide cutoff values for the proposed parameters(A-D-P-V) in the conclusion section from which adcerse cardiovascular or cerebrovascular events and end-organ hypertensive dammage are more likely?
Response: done. Thank you.
You wrote: An impressive number of studies conducted by this team is cited. Are all these self citations relevant and do they add to the value of this review?
Response: we carefully checked references and we removed some citations to reduce the rate of self-citations.
Round 2
Reviewer 1 Report
Great work!
Author Response
Thank you.
Reviewer 3 Report
Dear authors, you have adequately adressed most of my comments. Thank you! However, an impressive number of self citations persist.
Author Response
Thank you. We also reduced the number of self-citations.